# A Survey of Corpora for
# Germanic Low-Resource Languages and Dialects

**Verena Blaschke**          **Hinrich Schütze**          **Barbara Plank**

Center for Information and Language Processing (CIS), LMU Munich, Germany
Munich Center for Machine Learning (MCML), Munich, Germany

`blaschke@cis.lmu.de`    `inquiries@cislmu.org`    `bplank@cis.lmu.de`

## Abstract

Despite much progress in recent years, the vast majority of work in natural language processing (NLP) is on standard languages with many speakers. In this work, we instead focus on low-resource languages and in particular non-standardized low-resource languages. Even within branches of major language families, often considered well-researched, little is known about the extent and type of available resources and what the major NLP challenges are for these language varieties. The first step to address this situation is a systematic survey of available corpora (most importantly, annotated corpora, which are particularly valuable for NLP research). Focusing on Germanic low-resource language varieties, we provide such a survey in this paper. Except for geolocation (origin of speaker or document), we find that manually annotated linguistic resources are sparse and, if they exist, mostly cover morphosyntax. Despite this lack of resources, we observe that interest in this area is increasing: there is active development and a growing research community. To facilitate research, we make our overview of over 80 corpora publicly available.[1]

## 1 Introduction

The majority of current NLP today focuses on standard languages. Much work has been put forward in broadening the scope of NLP (Joshi et al., 2020), with long-term efforts pushing boundaries for language inclusion, for example in resource creation (e.g., Universal Dependencies (Zeman et al., 2022)) and cross-lingual transfer research (de Vries et al.,

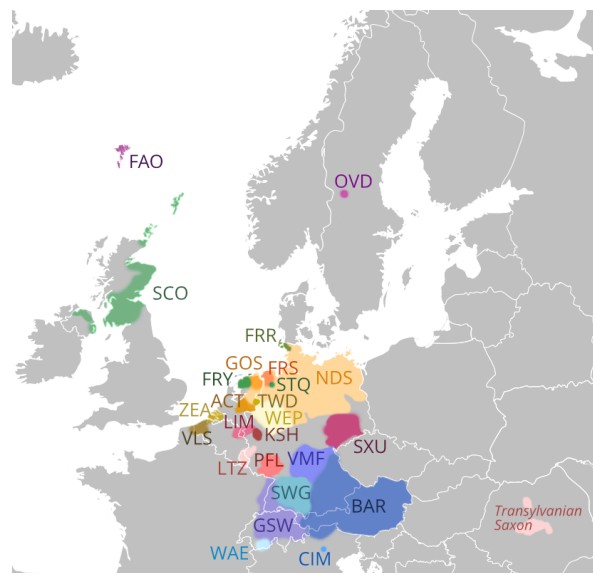

Figure 1: **Approximate locations of most of the languages discussed in this article** (not pictured: PDC, YID, NOR, SWE, DAN, ENG, DEU). Based on a map of Europe by Marian Sigler, CC BY-SA 3.0.

2022). However, even within major branches of language families or even single countries, plenty of language varieties are under-researched.

Current technology lacks methods to handle scarce data and the rich variability that comes with low-resource and non-standard languages. Nevertheless, interest in these under-resourced language varieties is growing. It is a topic of interest not only for (quantitative) dialectologists (Wieling and Nerbonne, 2015; Nerbonne et al., 2021), but also NLP researchers, as evidenced by specialized workshops like VarDial[2], special interest groups for endangered[3] and under-resourced languages,[4] and recent research on local languages spoken in Italy (Ramponi, 2022), Indonesia (Aji et al., 2022) and Australia (Bird, 2020), to name but a few.

---

[1]We share a companion website of this overview at `github.com/mainlp/germanic-lrl-corpora`.

[2]`sites.google.com/view/vardial-2023`
[3]SIGEL, `acl-sigel.github.io`
[4]SIGUL, `www.elra.info/en/sig/sigul`

In this paper, we provide an overview of the current state of NLP corpora for Germanic low-resource languages (LRLs) and dialects, with a particular focus on non-standard variants and four dimensions: annotation type, curation profile, resource size, and (written) data representation. We find that the amount and type of data varies by language, with manual annotations other than for morphosyntactic properties or the speaker's dialect or origin being especially rare. With this survey, we aim to support development of language technologies and quantitative dialectological analyses of Germanic low-resource languages and dialects, by making our results publicly available. Finally, based on the experiences we made while compiling this survey, we share recommendations for researchers releasing or using such datasets.

## 2 Related surveys

Zampieri et al. (2020) provide an overview on research on NLP for closely related language varieties and mention a few data sets. Recently, several surveys focusing on NLP tools and corpus linguistics data for regional languages and dialects have been released: for local languages in Italy (Ramponi, 2022) and France (Leixa et al., 2014), indigenous languages of the Americas (Mager et al., 2018), Arabic dialects (Shoufan and Alameri, 2015; Younes et al., 2020; Guellil et al., 2021), creole languages (Lent et al., 2022), Irish English (Vaughan and Clancy, 2016), and spoken varieties of Slavic languages (Dobrushina and Sokur, 2022). Furthermore, Bahr-Lamberti (2016) and Fischer and Limper (2019)[5] survey digital resources for studying varieties closely related to German, although these do not necessarily fit our inclusion criteria (cf. Section 4).

## 3 Language varieties

Our survey contains corpora for more than two dozen Germanic low-resource varieties, selected based on dataset availability (Appendix A contains the full list). We focus on specialized corpora showcasing regional variation, but not necessarily global variation. This overview does not include any corpora for Germanic-based creoles like Naija, as those are included in the recent survey by Lent et al. (2022). Figure 1 shows where most of the doculects included in this survey are spoken.

## 4 Methodology

Similarly to Ramponi (2022), we search for corpora on several online repositories for language resources: the CLARIN Virtual Language Observatory (Van Uytvanck et al., 2010), the LRE Map (Calzolari et al., 2012), the European Language Grid (Rehm et al., 2020) OLAC (Simons and Bird, 2003), ORTOLANG (Pierrel et al., 2017), and the Hamburg Centre for Language Corpora.[6] We also search for corpora on Zenodo and on Google Dataset Search, and look for mentions of corpora in articles hosted by the ACL Anthology and on ArXiv.[7] We search for mentions of the word "dialect" and the names of various Germanic low-resource languages.

We use the following inclusion criteria:

- The corpus is accessible to researchers (immediately via a website, or indirectly through a request form or via contact information),[8] and this is indicated on the corpus website or in a publication accompanying the corpus.

- The corpus can be downloaded easily (does not require scraping a website) and does not require extensive pre-processing (we are interested in file formats like XML, TSV or TXT rather than PDF).

- The data are of a high quality, e.g., we ignore OCR'ed corpora that were not carefully cleaned.

- The corpus (mainly) contains full sentences or utterances,[9] and the data were (mainly) produced in the past century.

We base this survey only on the versions of corpora that are easily accessible to the research community; e.g., if a corpus contains audio material, but only the written material is available for download (and thus as a data source for quantitative research), the corpus is treated as a text corpus.[10]

---

[5] regionalsprache.de/regionalsprachen forschung-online.aspx

[6] vlo.clarin.eu; lremap.elra.info; live.european-language-grid.eu; www.language-archives.org; www.ortolang.fr/market/corpora; corpora.uni-hamburg.de/hzsk/en/repository-search

[7] zenodo.org; datasetsearch.research.google.com; aclanthology.org; arxiv.org

[8] The latter case is indicated with a lock 🔒 in the tables.

[9] This excludes word lists and some heavily preprocessed corpora, like the one by Hovy and Purschke (2018), which is lemmatized and does not contain stop words.

[10] This is not a rare scenario, as the audio versions might

| Corpus | Langs | Annotation | Size | Rep. |
|---|---|---|---|---|
| UD Faroese OFT (Tyers et al., 2018) `github.com/UniversalDependencies/UD_Faroese-OFT` | FAO | POS (UPOS, Giellatekno-FAO), dep (UD), morpho (UD), lemmas | 1.2k sents | **A** |
| FarPaHC / UD Faroese FarPaHC (Ingason et al., 2012; Rögnvaldsson et al., 2012) `hdl.handle.net/20.500.12537/92` `github.com/UniversalDependencies/UD_Faroese-FarPaHC` | FAO | POS (mod. Penn-h, UPOS), phrase struc.(mod. Penn-h), dep (UD), morpho (UD) | 53k (FarP.) / 40k (UD.) toks | **A** |
| LIA Treebank / UD Norwegian NynorskLIA (Øvrelid et al., 2018) `tekstlab.uio.no/LIA/norsk/index_english.html` `github.com/UniversalDependencies/UD_Norwegian-NynorskLIA` `github.com/textlab/spoken_norwegian_resources/tree/master/treebanks/Norwegian-NynorskLIA` | NOR ♀ | POS (UPOS, mod. NDT), dep (UD, mod. NDT), lemmas, morpho (UD) | 77.7k toks (L.), 55k toks (UD) | ¶ ✑* |
| NDC Treebank (Kåsen et al., 2022; Johannessen et al., 2009) `tekstlab.uio.no/scandiasyn/download.html` `github.com/textlab/spoken_norwegian_resources/tree/master/treebanks/Norwegian-BokmaalNDC` | NOR ♀ | POS (mod. NDT), dep (mod. NDT), lemmas, morpho (mod. NDT) | 66k toks | ¶ ✑* |
| NorDial (subset) (Mæhlum et al., 2022) Contact authors 🔒 | NOR | POS (UPOS) | 35+ tweets | ✎ |
| POS-tagged Scots corpus (Lameris and Stymne, 2021) `github.com/Hfkml/pos-tagged-scots-corpus` | SCO | POS (UPOS) | 1k tokens | ✎/**A** |
| TwitterAAE-UD (Blodgett et al., 2016) `slanglab.cs.umass.edu/TwitterAAE` | ENG (AAVE) | Dep (UD) | 250 tweets | ✎ |
| UD Frisian/Dutch Fame (Braggaar and van der Goot, 2021; Yılmaz et al., 2016) `github.com/UniversalDependencies/UD_Frisian_Dutch-Fame` | FRY/NLD | POS (UPOS), dep (UD), code-switching | 400 sents | **A** |
| UD Low Saxon LSDC (Siewert et al., 2021) `github.com/UniversalDependencies/UD_Low_Saxon-LSDC` | NDS ♀ | POS (UPOS), dep (UD), morpho (UD), glosses (GML), lemmas | 95 sents | ✎ ¶* |
| Stemmen uit het verleden (annotated subset) (Lybaert et al., 2019; Van Keymeulen et al., 2019) `doi.org/10.18710/NSFN2B` | VLS ♀ | V2 variation | 1.4k sents | ✑ |
| Penn Parsed Corpus of Historical Yiddish (Santorini, 2021) `github.com/beatrice57/penn-parsed-corpus-of-historical-yiddish` | YID | POS (Penn-h), phrase struc. (Penn-h) | ca. 200k toks | * |
| Kontatto (Dal Negro and Ciccolone, 2020) `kontatti.projects.unibz.it` 🔒 | BAR (South Tyrol) | POS (unknown), lemmas (DEU) | 147k toks | 🎤✑ |
| Annotated Corpus for the Alsatian Dialects (Bernhard et al., 2018, 2019) `zenodo.org/record/2536041` | GSW (Alsatian) | POS (UPOS, mod. UPOS), lemmas, glosses (FRA) | 798 sents | ✎ |
| Bisame GSW (STIH, 2020; Millour and Fort, 2018) `hdl.handle.net/11403/bisame_gsw/v1` | GSW (Alsatian) | POS (mod. UPOS) | 382 sents | ✎ |
| Geparstes und grammatisch annotiertes Korpus schweizerdeutscher Spontansprachdaten (Schönenberger and Haeberli, 2019) (contact authors 🔒) | GSW (St. Gallen) | POS (mod. Penn-h), phrase struc. (Penn-h) | 100k+ toks | 🎤✑ |
| NOAH's corpus (Hollenstein and Aepli, 2015) `noe-eva.github.io/NOAH-Corpus` | GSW | POS (mod. STTS, subset: UPOS and STTS) | 115k toks | ✎ |
| UD Swiss German UZH (Aepli and Clematide, 2018) `github.com/UniversalDependencies/UD_Swiss_German-UZH` | GSW | POS (UPOS, mod. STTS), dep (UD) | 100 sents | ✎ |
| WUS_DIALOG_GSW (subset of *What's up, Switzerland?*) (Stark et al., 2014–2020; Ueberwasser and Stark, 2017) `whatsup.linguistik.uzh.ch` 🔒 | GSW ♀ | POS (mod. STTS) | 34.7k toks | ✎¶ |

Table 1: **Morphosyntactically annotated corpora.** Abbreviations for the annotation tag sets are explained in Section 5.1.1, as are the orthographies of entries with an asterisk (*). Other abbreviations and symbols: *Rep.* = 'data representation,' *dep* = 'syntactic dependencies,' *phrase struc* = 'phrase structure,' *morpho* = 'morphological features,' *mod.* = 'modified,' *AAVE* = 'African-American Vernacular English,' *GML* = 'Middle Low Saxon,' *NLD* = 'Dutch,' *FRA* = 'French,' 🔒 = access is not immediate, ♀ = fine-grained dialect distinctions, ✑ = phonetic/phonemic transcription, ✎ = pronunciation spelling, **A** = LRL orthography, ¶ = normalized orthography.

| Corpus | Langs | Annotation | Size | Rep. |
|---|---|---|---|---|
| TaPaCo (subset) (Scherrer, 2020) `zenodo.org/record/3707949` | NDS, GOS | paraphrases | 1107 sents (NDS), 122 sents (GOS) | ✎ |
| Wenkersätze (Wenker, 1889–1923; Schmidt et al., 2020–) `github.com/engsterhold/wenker-storage` | DEU* ⚲ | translations (across dialects, DEU) | 2210 samples×40 sents | ✏/✎ |
| SB-CH (subset) (Grubenmann et al., 2018) `github.com/spinningbytes/SB-CH` | GSW | sentiment | 2.8k sents | ✎ |
| SwissDial (Dogan-Schönberger et al., 2021) `projects.mtc.ethz.ch/swiss-voice-data-collection` 🔒 | GSW ⚲, WAE | topic, translations (across dialects and into DEU) | 2.5–4.6 hrs×8 lects | 🎤✎¶ |
| xSID/SID4LR (subset) (van der Goot et al., 2021; Aepli et al., 2023) `bitbucket.org/robvanderg/sid4lr` | GSW, BAR (South Tyrol) | slot and intent detection, translations (14 langs) | 800 sents | ✎ |

Table 2: **Corpora with semantic annotations or parallel sentences.** Abbreviations and symbols: *Rep.* = 'data representation,' 🔒 = access is not immediate, ⚲ = fine-grained dialect distinctions, 🎤 = audio, ✎ = pronunciation spelling, ¶ = standard orthography. *The Wenkersätze contain samples from various German dialects, but those are not annotated directly (only the town names are shared).

## 5 Corpora

Most of the language varieties we survey have no or only a very recent written tradition. This unique challenge is reflected in the different written formats used to represent the data (if the corpora contain any written material at all) and concerns both the transcription of audio data (Tagliamonte, 2007; Gaeta et al., 2022) as well as the elicitation of written data (Millour and Fort, 2020). We opted to discern between audio data 🎤 and the following written variants: standard orthographies (of the doculects themselves where existing **A** (e.g., West Frisian orthography), or of a closely related higher-resource language otherwise ¶), ad-hoc pronunciation spelling (by speakers of the doculect) ✎, and phonetic or phonemic transcriptions (by linguists) ✐. Appendix B provides examples.

The following corpora are sorted by annotation and curation type. For an overview sorted by language, see Appendix A. Some of the corpora share the same data sources. Appendix C lists the cases where we are aware of such overlaps.

### 5.1 Annotated corpora

This section only includes corpora with manual (or manually corrected) annotations.

### 5.1.1 Morphosyntactic annotation

Table 1 provides an overview of datasets with morphosyntactic annotations. These mostly contain

part-of-speech (POS) tags and/or syntactic dependencies. Such annotations are, for instance, of interest to dialectologists studying morphosyntactic variation (see for example Lybaert et al., 2019). Automatically generating high-quality morphosyntactic annotations for non-standard and/or low-resource data is not trivial, and the more annotated data are available for training, the better the results tend to be (Schulz and Ketschik, 2019; Scherrer et al., 2019a).

The annotation standards tend to either be general and cross-linguistically applicable (inviting comparisons between languages), or to be very specific to the language variety at hand. In the former case, annotations follow the guidelines from the Universal Dependencies project (Zeman et al., 2022) (UD, UPOS). In the latter case, tag sets created for a (usually closely related) higher-resource language are modified so that they capture the lower-resource language variety's idiosyncrasies. These specialized tag sets are based on: the annotations of the Giellatekno project (Wiechetek et al., 2022), the annotations developed for the Penn Parsed Corpora of Historical English (Penn-h),[11] the tag set of the Norwegian Dependency Treebank (NDT) (Solberg et al., 2014) (based on the Oslo-Bergen Tagger's tag set, OBT, (Johannessen et al., 2012)), and the Stuttgart-Tübingen tag set (STTS) (Schiller et al., 1999).

Most of the annotated corpora are presented only in one written form, typically either written in a standard orthography or pronunciation spelling.

---

contain more personally identifying information (like the voice of someone from a small speaker population), and it requires more work to censor locations or personal names in audio data than in text data (see also Seyfeddinipur et al., 2019).

[11] `ling.upenn.edu/hist-corpora/annotation/index.html`

| Corpus | Langs | Size | Rep. |
|---|---|---|---|
| Føroyskur talumálsbanki (Jacobsen, 2022) `clarino.uib.no/corpuscle-classic/corpus-list` 🔒 | FAO | 599.9k toks | **A** |
| BLARK 1.0 (Background text corpus) (Simonsen et al., 2022) (incl. FTS (Språkbanken and Fróðskaparsetur Føroya) and Faroese Korp (Giellatekno)) `maltokni.fo/en/resources` | FAO | 25M toks | **A** |
| Nordic Dialect Corpus (subset) (Johannessen et al., 2009) `tekstlab.uio.no/nota/scandiasyn` | NOR ♀, OVD ♀ | 1.9M toks (NOR), 15.7k toks (OVD) | ¶ (NOR: ☑) (OVD: **A**) |
| LIA Norsk (Øvrelid et al., 2018) `tekstlab.uio.no/LIA/korpus.html` | NOR ♀ | 3.5M toks | ☑ ¶ partially 🎤 |
| Talemålsundersøkelsen i Oslo (TAUS) (Tekstlab, 2020) `tekstlab.uio.no/nota/taus/` | NOR (East/West Oslo) ♀ | 388k toks | ☑ ¶ |
| NorDial (Barnes et al., 2021) (subset) `github.com/jerbarnes/nordial` | NOR | 348 tweets | ✏ |
| American Nordic Speech Corpus (CANS) (Johannessen, 2015) `tekstlab.uio.no/norskiamerika/korpus.html` | NOR (US/Canada) ♀, SWE (US) ♀ | 773k toks (NOR), 46k toks (SWE) | ☑ ¶ |
| Parallel dialectal–standard Swedish data (Hämäläinen et al., 2020; Ivars and Södergård, 2007) `zenodo.org/record/4060296` | SWE (Finland) ♀, | 86.5k tokens | ☑ ¶ |
| Danish Gigaword (subset) (Strømberg-Derczynski et al., 2021; Kjeldsen, 2019) `gigaword.dk` | DAN (Bornholm) | ca. 400k tokens | unk. |
| Scottish Corpus of Texts & Speech (SCOTS) (subset) (Anderson et al., 2007) `scottishcorpus.ac.uk` | SCO | (unknown how many of 4.6M toks in SCO) | mix of ✏ ¶ |
| Low Saxon Dialect Classification (LSDC) (Siewert et al., 2020) `github.com/Helsinki-NLP/LSDC/` | NDS, WEP, FRS, TWD, ACT ♀ | 88.9k sents | ✏ |
| LuxId (Lavergne et al., 2014) `lrec2014.lrec-conf.org/en/shared-lrs/current-list-shared-lrs` | LTZ/DEU/FRA code-switching | 924 sents (most with LTZ content) | **A** |
| DiDi (subset) (Frey et al., 2019) `hdl.handle.net/20.500.12124/7` | BAR (South Tyrol) | unknown | ✏ |
| What's up, Switzerland? (Stark et al., 2014–2020; Ueberwasser and Stark, 2017) `whatsup.linguistik.uzh.ch` 🔒 | GSW ♀ | 507k msgs / 3.6M toks | ✏ |
| Swatchgroup Geschäftsbericht (subset) (Graën et al., 2019) `pub.cl.uzh.ch/wiki/public/pacoco/start` | GSW | 79.6k toks | ✏ |
| Text+Berg (subset) (Bubenhofer et al., 2015; Graën et al., 2019) `textberg.ch/site/en/corpora` 🔒 `pub.cl.uzh.ch/wiki/public/pacoco/start` | GSW | 156 sents / 3.1k toks | ✏ |
| ArchiWals / CLiMAlp (Angster et al., 2017; Gaeta, 2020) `climalp.org` 🔒 | WAE ♀ | 80+k tokens | ✏ |

Table 3: **Other curated text corpora.** Abbreviations and symbols: *Rep.* = 'data representation,' 🔒 = access is not immediate, ♀ = fine-grained dialect distinctions, ☑ = phonetic/phonemic transcription, ✏ = pronunciation spelling, **A** = LRL orthography, ¶ = normalized orthography.

| Corpus | Langs | Size | Rep. |
|---|---|---|---|
| BLARK 1.0 (Transcr. recordings) (Simonsen et al., 2022) `maltokni.fo/en/resources` | FAO 📍 | 100 h | 🎙 **A** (some ✏️) |
| Faroese Danish Corpus Hamburg (FADAC Hamburg) (subset) (Debess, 2019) `corpora.uni-hamburg.de/hzsk/de/islandora/object/spoken-corpus:fadac-0.2.dan` | FAO 📍 | 31 h | 🎙 **A** |
| NB Tale – Speech Database for Norwegian (Språkbanken) `nb.no/sprakbanken/en/resource-catalogue/oai-nb-no-sbr-31/` | NOR 📍 | 365 × 2 min (spon.), 7.6k sents (reading) | 🎙 ✏️ ¶ |
| Norwegian Parliamentary Speech Corpus (NPSC) (Solberg and Ortiz, 2022) `nb.no/sprakbanken/en/resource-catalogue/oai-nb-no-sbr-58/` | NOR 📍 | 140 h | 🎙 ¶ |
| Diachronic Electronic Corpus of Tyneside English (DECTE) (Corrigan et al., 2012) `research.ncl.ac.uk/decte/index.htm` 🔒 | ENG (UK: Tyneside) | 72 h / 804k toks | 🎙 ¶ (some ✏️) |
| Intonational Variation in English (IViE) (Nolan and Post, 2014) `phon.ox.ac.uk/files/apps/IViE/` | ENG (UK, Ireland) 📍 | 36 h | 🎙 ¶ |
| Crowdsourced high-quality UK and Ireland English Dialect speech data set (Demirsahin et al., 2020) `openslr.org/83` | ENG (UK, Ireland) 📍 | 31 h | 🎙 ¶ |
| Helsinki Corpus of British English Dialects (University of Helsinki, 2006) `varieng.helsinki.fi/CoRD/corpora/Dialects/` 🔒 | ENG (UK) 📍 | 1 M toks | 🎙 ¶ |
| Nationwide Speech Project (NSP) (Clopper and Pisoni, 2006) `u.osu.edu/nspcorpus` | ENG (US) 📍 | 60 × 1 hr | 🎙 (some ¶) |
| Corpus of Regional African American Language (CORAAL) (Kendall and Farrington, 2021) `oraal.uoregon.edu/coraal` | ENG (AAVE) 📍 | 135.6 hrs / 1.5M toks | 🎙 ¶ |
| Common Voice Corpus 12.0 (subset) (Ardila et al., 2020) `commonvoice.mozilla.org/en/datasets` | FRY | 150 h | 🎙 **A** |
| Frisian AudioMining Enterprise (FAME) (Yılmaz et al., 2016) `ru.nl/clst/tools-demos/datasets/` 🔒 | FRY (some 📍) | 18.5 h | 🎙 **A** |
| Recordings of Dutch-Frisian council meetings (Bentum et al., 2022) `frisian.eu/dutchfrisiancouncilmeetings` | FRY | 26 h / 281k toks | 🎙 **A** |
| Corpus Spoken Frisian (Frisian Academy) `www1.fa.knaw.nl/ksf.html` 🔒 | FRY | 200 h (65 h transcribed) | 🎙 (**A**) |
| Sprachvariation in Norddeutschland (SiN, Hamburg collection) (Schröder, 2011; Elmentaler et al., 2015) `hdl.handle.net/11022/0000-0000-7EE3-3` 🔒 | NDS, FRS, DEU | 300 h | 🎙 |
| Regional Variants of German 1 (RVG1) (Burger and Schiel, 1998) `hdl.handle.net/11022/1009-0000-0004-3FF4-3` | DEU* 📍 | 500+ × 1 min | 🎙 ✏️ ¶ |
| Zwirner-Korpus (downloadable subset) (Zwirner and Bethge, 1958; IDS) `dgd.ids-mannheim.de` 🔒 | NDS, WEP, SXU, VMF, BAR, GSW, DEU** 📍 | 3 h / 24.8k toks | 🎙 ¶ |
| Texas German Sample Corpus (TGSC) (Blevins, 2022) `doi.org/10.18738/T8/IOX9ZA` | DEU (Texas) | 13.5 h / 75k toks | 🎙 ¶ |
| Audioatlas Siebenbürgisch-Sächsischer Dialekte (University of Munich) `hdl.handle.net/11022/1009-0000-0001-27B9-3` 🔒 | DEU (Trans. Saxon)*** | 360 h / 450k toks | 🎙 ¶ (some ✏️) |
| CABank Yiddish Corpus (Newman, 2015) `ca.talkbank.org/access/Yiddish.html` | YID (New York) | 1 hr | 🎙 ✏️ |
| SXUCorpus (Herms et al., 2016) Contact authors 🔒 | SXU 📍 | 500 min / 70k toks | 🎙 ¶ |
| Kontatti (Ghilardi, 2019) `kontatti.projects.unibz.it` 🔒 | BAR (S. Tyrol), CIM | unknown | 🎙 ¶ |
| ArchiMob (Scherrer et al., 2019b) `spur.uzh.ch/en/departments/research/textgroup/ArchiMob.html` (audio files: 🔒) | GSW 📍 | 70 h | 🎙 ✏️ ¶ |
| SDS-200 (Plüss et al., 2022) `swissnlp.org/datasets/` 🔒 | GSW | 200 h | 🎙 ¶ |
| Swiss Parliaments Corpus (Plüss et al., 2021a) `www.cs.technik.fhnw.ch/i4ds-datasets` | GSW | 293 h | 🎙 ¶ |
| Gemeinderat Zürich Audio Corpus (Plüss et al., 2021b) `www.cs.technik.fhnw.ch/i4ds-datasets` | GSW | 1208 h | 🎙 |
| All Swiss German Dialects Test Set (Plüss et al., 2021b) `www.cs.technik.fhnw.ch/i4ds-datasets` | GSW, WAE 📍 | 13 h / 5.8k utterances | 🎙 ¶ |
| Walliserdeutsch/RRO (Garner, 2014; Garner et al., 2014) `zenodo.org/record/4580286` 🔒 | WAE | 8.3 h | 🎙 ✏️ |

Table 4: **Other audio corpora.** Abbreviations and symbols: *Rep.* = 'data representation,' 🔒 = access is not immediate, 📍 = fine-grained dialect distinctions, 🎙 = audio, ✏️ = phonetic/phonemic transcription, ✏️ = pronunciation spelling, **A** = LRL orthography, ¶ = normalized orthography. *It is unclear whether the RVG1 recordings are in regionally accented (Standard) German or whether they are in Low Saxon, Bavarian and other regional languages spoken in Germany, Switzerland, Austria and Northern Italy. **The Zwirner-Korpus contains samples from various dialects spoken in what used to be West Germany. ***Transylvanian Saxon is a variety of Moselle Franconian that does not have its own ISO code. It is more closely related to Luxembourgish than to Standard German.

| Corpus | Languages and sizes |
|---|---|
| Tatoeba (subset; with > 100 sents) `tatoeba.org/en/downloads` | in sentences: NDS (18.1k), YDD (12.8k), GOS (5.7k), FRR (2.9k), SWG (1.9k), LTZ (884), FRY (641), GSW (474), FAO (417), BAR (227) |
| Ubuntu `opus.nlpl.eu/Ubuntu.php` | in toks: NDS (35.3k), FRY (22.4k), FAO (20.2k), LIM (18.4k), LTZ (17.0k) |
| KDE4 `opus.nlpl.eu/KDE4-v2.php` | NDS (1.1M toks), FRY (0.3M toks), LTZ (28.8k toks) |
| GNOME `opus.nlpl.eu/GNOME.php` | NDS (0.7M toks), LIM (0.4M toks), FRY (55.7k toks) |
| Mozilla-I10n `mozilla-l10n/mt-training-data` | FRY (0.4M toks), LTZ (6.9k toks) |
| QED (Abdelali et al., 2014) `opus.nlpl.eu/QED.php` | LTZ (19.2k toks), FAO (6.4k toks) |
| TED2020 (Reimers and Gurevych, 2020) `opus.nlpl.eu/TED2020.php` | LTZ (1.7k toks) |
| Danish Gigaword (subset) (Strømberg-Derczynski et al., 2021) `gigaword.dk` | DAN (South Jutish) (ca. 20k tokens) |
| SwissCrawl (Linder et al., 2020) `icosys.ch/swisscrawl` 🔒 | GSW (500k+ sents) |
| SB-CH (Grubenmann et al., 2018) `github.com/spinningbytes/SB-CH` 🔒 | GSW (ca. 116k sents) |
| SwigSpot (Linder, 2018) `github.com/derlin/SwigSpot_Schwyzertuutsch-Spotting` | GSW (8k sents) |
| Web to Corpus (W2C) (subset) (Majliš, 2011; Majliš and Žabokrtský, 2012) `hdl.handle.net/11858/00-097C-0000-0022-6133-9` | in MB: YID (125), FAO (102), LTZ (81), FRY (72), SCO (35), NDS (24), LI (20) |
| CC-100 (subset) (Wenzek et al., 2020) `data.statmt.org/cc-100/` | FRY (174 MB), YID (51 MB), LIM (8.3 MB) |
| OSCAR (subset) (Abadji et al., 2022) `oscar-project.github.io/documentation/` 🔒 | in toks: YID (14.3M), FRY (9.8M), LTZ (2.5M), NDS (1.6M), GSW (34k) |
| Wikipedia (subset) `dumps.wikimedia.org` | discussed in detail in Appendix D |

Table 5: **Uncurated corpora.** 🔒 = Access not immediate. The corpora in the top section contain parallel sentences with many translations and are (also) distributed via the OPUS project (Tiedemann, 2012).

Some cases (marked with an asterisk* in the table) require further explanation: The Norwegian LIA and NDC treebanks (Øvrelid et al., 2018; Kåsen et al., 2022) use normalized orthographies (Nynorsk and Bokmål, respectively), but aligned versions of the original phonetic and orthographic transcriptions can be downloaded from the Tekstlab links in the table. The sentences in the UD Low Saxon LSDC treebank (Siewert et al., 2021) are presented both in the original ad-hoc pronunciation spelling and in a recently proposed orthography for Low Saxon, *Nysassiske Skryvwyse*. The Yiddish corpus (Santorini, 2021) is romanized, partially according to the YIVO transliteration system, and partially in a non-systematic manner.

### 5.1.2 Semantic annotation and parallel sentences

Very few resources with other types of annotations exist; we were able to find only five (Table 2), all of which have very different kinds of annotations: sentiment or topic classification, intent detection and slot-filling, translations and paraphrases.

### 5.1.3 Dialect annotation

Many corpora contain detailed annotations on the dialect area (or more precise location) an utterance's speaker or the author of a document is from. Such information is important for linguistic research comparing related dialects (Wieling and Nerbonne, 2015), for comparing the results of traditional and quantitative dialectological approaches (e.g. Heeringa et al., 2009) and for evaluating whether NLP systems perform differently on different closely related language varieties (Ziems et al., 2022). Since corpora with such annotations belong to all of the categories of curated datasets in this survey, they are not presented on their own, but instead marked with a pin symbol 📍 elsewhere.

### 5.2 Other curated corpora

### 5.2.1 Text corpora

Table 3 presents unannotated written corpora of low-resource languages like Elfdalian or Faroese, and corpora that showcase dialectal variation through phonetic transcriptions or pronunciation spelling. (While variation also occurs on linguistic levels encoded in normalized text written in stan-

dard orthographies – lexical, syntactic or pragmatic variation – we focus on phonological variation, as this is where specialized corpora are required.)

### 5.2.2 Audio corpora

In this survey, our focus lies on written resources, and as such, this selection of audio corpora is not exhaustive.[12] However, many of the language varieties surveyed in this article are predominately spoken rather than written. Creating language technology for unwritten languages is a topic of interest for NLP researchers (Scharenborg et al., 2020), and this is also reflected by the number of recently created speech corpora for Germanic LRLs.

Many of the audio corpora (Table 4) fall into one of two categories: recordings created for dialectological research, and post-hoc collections of already existing audio data (like radio broadcasts or public recordings of council meetings). Most of the audio corpora are at least partially transcribed, typically according to a standard orthography.

### 5.3 Uncurated text corpora

A final type of corpus are uncurated text collections (Table 5). This includes data coming from community-based data collection efforts unrelated to research projects (Wikipedia, Tatoeba) and open-source translations of (mostly) user interfaces, as well as web-crawled data.

It is important to note that there are quality issues with web-crawled corpora, especially for low-resource languages (Kreutzer et al., 2022).[13] Both CC-100 (Wenzek et al., 2020) and OSCAR (Abadji et al., 2022) are cleaned versions of CommonCrawl[14]– and Abadji et al. (2022) specifically remark on the low quality of the low-resource language data in that dataset.

Some of the translated corpora also have quality issues: the Low Saxon Ubuntu and GNOME corpora (Tiedemann, 2012) both also contain some Standard German content. We exclude subcorpora that contain mostly foreign language or non-linguistic material (for instance, the West Flemish QED subcorpus (Abdelali et al., 2014; Tiedemann, 2012)).

Wikipedia has editions in many Germanic low-resource languages and at different activity and contributor levels, as we survey in Appendix D. Projects extend wiki dumps with automatically inferred annotations (Pan et al., 2017; Schwenk et al., 2021), or release automatically aligned German–Alemannic/Bavarian bitext (Artemova and Plank, 2023).[15] The linguistic quality of LRL wikis is not always very high – the Scots Wikipedia made the news in 2020, when attention was brought to the fact that half of that wiki's articles had been created/edited by a non-Scots speaker writing in a parody of Scots (Brooks and Hern, 2020). Quality issues should be taken into account when working with data from small wikis without much oversight, e.g., with data or tools based on the Scots Wikipedia before clean-up started in fall 2020.[16]

## 6 Outlook

Creating NLP resources and technology for LRLs is an active field. At the time of writing this paper, several additional resources were concurrently under construction or revision: *UD Frisian Frysk*, a treebank for West Frisian (Heeringa et al., 2021),[17] *Boarnsterhim Corpus*, a West Frisian audio corpus (Sloos et al., 2018),[18] *Schweizerdeutsches Mundartkorpus*, a Swiss German text corpus (Weibel and Peter, 2020),[19] and the *Corpus of Southern Dutch Dialects* (Breitbarth et al., 2018).[20] Community-based projects are also being actively developed: many of the small Wikipedias have active editors (Appendix D), as do many of the Tatoeba collections. We welcome contributions to our companion website to track such progress.

Speaker populations of LRLs are not a monolith. Accordingly, different speaker communities have different interests in terms when it comes to the development of language technologies (Lent et al., 2022). The creation of downstream technologies made for public use should be made in accordance of the wishes and needs of the relevant speaker communities (see also Bird, 2022).

---

[12]Additional corpora documenting variation in spoken English can be found via the SPADE project (Stuart-Smith et al., 2017-2020).

[13]However, see Artetxe et al. (2022) for an argument that the linguistic quality of a corpus might not be the most important factor for all downstream applications.

[14]`commoncrawl.org`

[15]`github.com/mainlp/dialect-BLI`

[16]E.g., Scots is included in the language list of mBERT (Devlin et al., 2019), which was trained on Wikipedia data in 2019: `github.com/google-research/bert/blob/master/multilingual.md`

[17]`github.com/UniversalDependencies/UD_Frisian-Frysk`

[18]`taalmaterialen.ivdnt.org/download/tstc-boarnsterhimcorpus1-0`

[19]`chmk.ch/de/info_all`

[20]`gcnd.ugent.be`

We make the following **recommendations** for researchers who *work* with LRL datasets:

- Investigate the quality of uncurated data, as it might be especially poor for LRLs.

- Check whether (pre-)training, development and test data are truly from independent datasets – the dearth of high-quality LRL data means that datasets may be likely to overlap.

- Consider quantitative work by dialectologists and sociolinguists who might not publish in typical NLP venues.

To researchers who *create* such datasets, we recommend to:

- Document the transcription principles (if the data were originally in an audio format) / if any standardized orthographies were used (if the language variety does not have an official orthography).

- The low number of available high-quality datasets per language variety means that the impact of losing such a resource is much greater. Therefore, please upload your corpus to an archive geared towards long-term data storage (like the CLARIN Language Resource Inventory,[21] the LRE Map or Zenodo).

- Provide easy-to-find documentation with details on the corpus size, data sources and the annotation procedure.

## 7 Conclusion

We have presented an analysis of over 80 corpora containing data in Germanic low-resource languages, with a focus on non-standardized or only recently standardized varieties. We additionally share the corpus overview on a public companion website (`github.com/mainlp/germanic-lrl-corpora`) that can easily be updated as more language resources are released.

## Acknowledgements

We thank the anonymous reviewers as well as the members of the MaiNLP research lab for their constructive feedback. This research is supported by European Research Council (ERC) Consolidator Grant DIALECT 101043235. This work was partially funded by the ERC under the European Union's Horizon 2020 research and innovation program (grant 740516).

---

[21]`clarin.eu/content/language-resource-inventory`

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

## A Resources by language

We include the languages associated with the ISO 639-3 codes FAO (Faroese), OVD (Elfdalian), SCO (Scots), FRR (North Frisian), FRY (West Frisian), STQ (Saterland Frisian), NDS (Low Saxon), FRS (East Frisian Low Saxon), GOS (Gronings), TWD (Twents), ACT (Achterhoeks), WEP (Westphalian), ZEA (Zeelandic), VLS (West Flemish), LTZ (Luxembourgish), LIM (Limburgish), KSH (Colognian), PFL (Palatine German), PDC (Pennsylvania Dutch), YID (Yiddish), SXU (Upper Saxon), VMF (East Franconian), BAR (Bavarian), SWG (Swabian), GSW (Swiss German and Alsatian), WAE (Walser), and CIM (Cimbrian). Our survey also encompasses data for dialects/non-standard varieties of Norwegian (NOR), Swedish (SWE), Danish (DAN), English (ENG), and German (DEU) that do not have their own ISO codes.

We use ISO codes to refer to (groups of) language varieties for practical reasons – despite their shortcomings as labels for varieties from linguistic continua (Morey et al., 2013; Nordhoff and Hammarström, 2011), they are widely used and recognized, and many of the corpora in this survey are described in terms that easily map to ISO codes.

In some cases, the codes or the corpus descriptions are ambiguous. For instance, many Low Saxon corpora contain entries that also belong to one of the more specific Dutch Low Saxon codes, and some Swiss German corpora also contain some Walser content. Where possible (and where the data instances themselves are labelled on a precise enough level), we use the more specific codes.

Table 6 provides an overview of resource types by language variety.

## B Written representations

Table 7 provides examples of different types of written representations and showcases how diverse each category can be.

Examples 1a, 2a, 3a, 4a/b, 5a and 6a are written in **standardized orthographies** (or in lower-cased versions of standard orthographies with no pronunciation). Of these, sentences 1a, 4a and 5a are written in orthographies developed for their respective low-resource languages **A**, while 2a, 3a, 4b and 6a are normalized and written in the orthographies of closely related standard languages **¶** (the last two are Elfdalian written in Swedish and Swiss German written in Standard German).

Sentences 5b and 7a present two examples of ad-hoc **pronunciation spellings ✏**. These kinds of spellings vary from speaker to speaker, and one and the same speaker might also choose different spellings of the same word at different times.

**Phonetic or phonemic transcriptions ✐** have

| Language | | Dialect/ Location | Morpho-syntax | Semantic | Parallel (curated) | Uncurated text | Curated data |
|---|---|---|---|---|---|---|---|
| *North Germanic* | | | | | | | |
| FAO | Faroese | 📍 | ✔ | | | ✔ | 🎤 ⊠ **A** |
| NOR | (non-std.) Norwegian | 📍 | ✔ | | | | 🎤 ⊠ ✎ ¶ |
| OVD | Elfdalian | 📍 | | | | | **A** ¶ |
| SWE | (non-std.) Swedish | 📍 | | | | | ⊠ ¶ |
| DAN | (non-std.) Danish | 📍 | | | | ✔ | ? |
| *Anglo-Frisian* | | | | | | | |
| SCO | Scots | | ✔ | | | ✔ | ✎ **A** ¶ |
| ENG | (non-std.) English | 📍 | ✔ | | | | 🎤 ¶ |
| FRY | West Frisian | 📍 | ✔ | | | ✔ | 🎤 **A** |
| FRR | North Frisian | | | | | ✔ | |
| STQ | Saterland Frisian | | | | | ✔ | |
| *Low German\** | | | | | | | |
| NDS | Low Saxon | 📍 | ✔ | | ✔ | ✔ | 🎤 ✎ **A** |
| FRS | East Frisian Low Saxon | | | | | ✔ | 🎤 |
| GOS | Gronings | | | | ✔ | ✔ | |
| TWD | Twents | | | | | ✔ | ✎ |
| ACT | Achterhoeks | | | | | ✔ | ✎ |
| WEP | Westphalian | | | | | | 🎤 ¶ |
| *Macro-Dutch* | | | | | | | |
| VLS | West Flemish | 📍 | ✔ | | | ✔ | ⊠ |
| ZEA | Zeelandic | | | | | ✔ | |
| *Middle German* | | | | | | | |
| LTZ | Luxembourgish | | | | | ✔ | **A** |
| KSH | Colognian | | | | | ✔ | |
| LIM | Limburgish | | | | | ✔ | |
| PFL | Palatine German | | | | | ✔ | |
| PDC | Pennsylvania Dutch | | | | | ✔ | |
| YID | Yiddish\*\* | | ✔ | | | ✔ | 🎤 ⊠ |
| SXU | Upper Saxon | | | | | | 🎤 ¶ |
| *Upper German* | | | | | | | |
| DEU | (non-std.) German | | | | ✔ | | 🎤 ⊠ ¶ |
| VMF | East Franconian | | | | | | 🎤 ¶ |
| BAR | Bavarian | | ✔ | ✔ | ✔ | ✔ | 🎤 ⊠ ✎ ¶ |
| CIM | Cimbrian | | | | | | 🎤 ¶ |
| SWG | Swabian | | | | | ✔ | |
| GSW | Swiss Ger. & Alsatian | 📍 | ✔ | ✔ | ✔ | ✔ | 🎤 ⊠ ✎ ¶ |
| WAE | Walser | 📍 | | ✔ | ✔ | ✔ | 🎤 ✎ |

Table 6: **Corpora by language variety.** For ease of reference, the language are sorted by Germanic subbranches (based on Glottolog (Hammarström et al., 2022)). *For additional texts written in varieties of Low German/Saxon with other ISO 639-3 codes, see the note on the Low Saxon Wikipedias in Table 8. **Glottolog discerns between Eastern Yiddish (Middle German) and Western Yiddish (Upper German). Symbols: 🎤 = audio, ⊠ = phonetic/phonemic transcription, ✎ = pronunciation spelling, **A** = LRL orthography, ¶ = normalized orthography.

From the Faroese BLARK recordings (Simonsen et al., 2022):

1a **A** vit høvdu matpakka við og eg hugnaði mær óført

1b ✎ vId h9dI m%EApaHga v%i: o e h%u:najI mar %OW:f9zd

1c ✎ vɪd hœdɪ ˈmɛa̯pʰaʰga ˈviː o e ˈhuːnajɪ maɹ ˈɔu̯ːfœ̣sd

"We had lunchboxes with us and I enjoyed myself greatly."

From the Norwegian NB Tale corpus (Språkbanken):

2a ¶ Etter litt godsnakk kom tre av kyrne mot han mens den fjerde glei og fall

2b ✎ ""{t@4 l"it g""u:snAkk k"Om t4"e: "A:v C"y:n'@ m"u:t "An m"ens d_= fj""{:d'@ gl"eI "O: f"Al

2c ✎ ²ɛtəɹ ¹lɪt ²guːsnɑkk ¹kɔm ¹tɹeː ¹ɑːv ¹çyːŋə ¹muːt ¹ɑn ¹mɛns dn̩ ²fjæːɖə ¹glɛɪ̯ ¹oː ¹fɑl

"After some coaxing, three of the cows came towards him while the fourth one slipped and fell."

From the Norwegian part of the Nordic Dialect Corpus (Johannessen et al., 2009):

3a ¶ det er slik at de fleste kommer jo att når de får # unger

3b ✎ de e sjlik att dæi fLeste kjemme jo att nårr dæi fær # onnga

"The thing is that most people return when they have *[brief pause]* kids."

From the Elfdalian part of the Nordic Dialect Corpus (Johannessen et al., 2009):

4a **A** wen wa wen war eð før ien månað ? juni ?

4b ¶ vad va- vad var det för en månad ? juni ?

"What, wha-, what month was it? June?"

From UD Low Saxon LSDC (Siewert et al., 2021):

5a **A** Nu leyt em de böyse vynd disse nacht kyn ouge an enander doon.

5b ✐ Nu leit em de baise Find düse Nacht kinn Auge an enander dohn.

"Now the wicked enemy didn't let them get a wink of sleep that night."

From the Swiss German ArchiMob corpus (Scherrer et al., 2019b):

6a ¶ können sie ihre jugendzeit beschreiben

6b ✎ chönd sii iri jugendziit beschriibe

"Can you describe your youth?"

From the BISAME corpus (STIH, 2020):

7a ✐ Niema hat salamols gweßt as die Werter vum franzeescha kumma.

"Nobody knew then that these words came from French."

Table 7: **Examples of written representations.** Symbols: ✎ = phonetic/phonemic transcription, ✐ = pronunciation spelling, **A** = LRL orthography, ¶ = normalized orthography.

different styles depending on each corpus's transcription guidelines. Examples 1b and 2b are written in modified versions of SAMPA and X-SAMPA, and the corpora come with sufficient documentation to automatically convert these transcriptions into IPA (1c, 2c). (The superscript symbols [1] and [2] in example 2c are commonly used to indicate the Norwegian pitch accent.) The transcription styles presented in examples 3b and 6b are based on Norwegian and Standard German orthography, respectively. What sets them apart from pronunciation spellings is that they are consistent across the entire corpus and that they follow linguistic rationales that often are outlined in the corpus documentation.

## C  Overlapping data sources

Several of the corpora mentioned in this article overlap with each other:

- *UD Faroese OFT* and the *Korp* subcorpus of the background corpus of the Faroese *BLARK 1.0* contain material from the Faroese Wikipedia.

- The *NDC Treebank* uses data from the *Nordic Dialect Corpus.*

- The *LIA Treebank* and *UD Norwegian NynorskLIA* are annotated subsets of *LIA Norsk*, and they overlap with each other.

- The *POS-tagged Scots corpus* contains annotated sentences from *SCOTS.*

- *UD Low Saxon LSDC* and *LSDC* overlap.

- *UD Frisian/Dutch Fame* is an annotated subset if *FAME.*

- Many of the sentences in *UD Swiss German UZH* are also in *NOAH's corpus.* Both of these corpora contain material from the Alemannic Wikipedia.

- *SB-CH* contains *NOAH's corpus.*

- The *Annotated Corpus for the Alsatian Dialects* contains articles from the Alemannic Wikipedia that were explicitly tagged as Alsatian.

- *TaPaCo* is a subset of *Tatoeba.*

- Any corpus that includes data from the internet might overlap with the uncurated datasets in Section 5.3.

## D  Wikipedia statistics

Table 8 provides a comparison of Wikipedia sizes and user (vs. bot) activity.[22] The sizes of the small Germanic Wikipedias vary considerably from wiki to wiki (there are just under 2k Pennsylvania Dutch articles, while the (German) Low Saxon Wikipedia has over 84k articles), as does the number of recently active contributors (from 6 active non-bot users per month for Ripurarian/Colognian, Palatine German and Pennsylvania Dutch to 70 for Scots).

While bots can be used for automating many tasks that are unrelated to the textual diversity of a wiki (e.g., cleaning up article redirection pages), they can also be used to automatically create short template-based articles.[23] The share of manual edits (i.e., edits not by bots) is very varied across wikis – only about a quarter of all edits in the Pennsylvania Dutch Wikipedia have been made manually, compared to 79 % in the North Frisian Wikipedia. However, there is a clear trend towards a much larger proportion of manual edits: the vast majority of edits made only in the past year were manual edits.

Some of the wikis are written according to one or more orthographies, while others either do not include any spelling recommendations at all or encourage editors to use whatever pronunciation spelling they prefer. The Dutch Low Saxon Wikipedia, for instance, recommends *Nysassiske Skryvwyse*, whereas the German Low Saxon Wikipedia recommends another orthography: *Sass-Schrievwies*. The Ripurarian/Colognian Wikipedia, conversely, encourages idiosyncratic spellings.[24]

---

[22]The data sources are the automatically updated list of Wikipedia sizes at `meta.wikimedia.org/wiki/List_of_Wikipedias_by_language_group#Germanic` (last accessed 2023-01-31) and Wikimedia's metadata via `wikimedia.org/api/rest_v1/`. The scripts are available via `github.com/mainlp/wikistats`.

[23]For an example for the latter, see `nds.wikipedia.org/wiki/Bruker:ArtikelBot`

[24]These are the pages detailing orthographic conventions we were able to find (sorted by wiki size): `nds.wikipedia.org/wiki/Wikipedia:Sass`; `sco.wikipedia.org/wiki/Wikipedia:Spellin_an_grammar`; `als.wikipedia.org/wiki/Hilfe:Schrybig`; `bar.wikipedia.org/wiki/Wikipedia:Wia_schreib_i_a_guads_Boarisch%3F`; `frr.wikipedia.org/wiki/Wikipedia:Spräkekoordinasjoon`; `li.wikipedia.org/wiki/Wikipedia:Wie_sjrief_ich_Limburgs`; `vls.wikipedia.org/wiki/Wikipedia:Gebruuk_van_streektoaln`; `nds-nl.wikipedia.org/wiki/Wikipedia:Spelling`; `stq.wikipedia.org/wiki/`;

Several of these wikis include (some) articles with metadata specifying which variety the document is written in.[25]

––––––––––––––
`Wikipedia:Hälpe_bie_ju_seelter_Sproake;`
`ksh.wikipedia.org/wiki/Wikipedia:`
`Schrievwies`

[25]Sorted by wiki size: `nds.wikipedia.org/wiki/`
`Kategorie:Artikels_na_Dialekt;als.`
`wikipedia.org/wiki/Kategorie:Wikipedia:`
`Dialekt;bar.wikipedia.org/wiki/Kategorie:`
`Artikel_nach_Dialekt;frr.wikipedia.org/`
`wiki/Kategorie:Spriakwiisen;li.wikipedia.`
`org/wiki/Categorie:Wikipedia:Artikele_`
`nao_dialek;vls.wikipedia.org/wiki/`
`Categorie:Wikipedia:Artikels_noar_`
`dialect;nds-nl.wikipedia.org/wiki/`
`Kategorie:Nedersaksies_artikel;ksh.`
`wikipedia.org/wiki/Saachjrupp:Wikipedia:`
`Atikkel_ier_Shprooche;pfl.wikipedia.org/`
`wiki/Sachgrubb:Adiggel_noch_em_Dialegd`

| Wikipedia & Language | | Articles (01/2023) | Manual edits (2001–2022) | Manual edits (2022) | Monthly editors (2022) |
|---|---|---|---|---|---|
| nds | NDS (Germany)* (📍) | 84 k | 44 % | 99 % | 30 |
| lb | LTZ | 61 k | 43 % | 85 % | 56 |
| fy | FRY | 50 k | 60 % | 99 % | 54 |
| sco | SCO | 39 k | 53 % | 63 % | 70 |
| als | GSW + SWG + WAE (📍) | 30 k | 69 % | 100 % | 58 |
| bar | BAR (📍) | 27 k | 68 % | 63 % | 39 |
| frr | FRR (📍) | 17 k | 79 % | 85 % | 16 |
| yi | YID | 15 k | 49 % | 97 % | 35 |
| li | LIM | 14 k | 42 % | 75 % | 21 |
| fo | FAO | 14 k | 41 % | 99 % | 29 |
| vls | VLS (📍) | 8 k | 45 % | 79 % | 16 |
| nds-nl | NDS (Netherlands)* (📍) | 8 k | 40 % | 68 % | 14 |
| zea | ZEA | 6 k | 47 % | 98 % | 10 |
| stq | STQ | 4 k | 38 % | 81 % | 8 |
| ksh | KSH + other Ripuarian (📍) | 3 k | 32 % | 99 % | 6 |
| pfl | PFL + oth. Rhen. Franc., Hessian (📍) | 3 k | 65 % | 72 % | 6 |
| pdc | PDC | 2 k | 27 % | 92 % | 6 |
| en | ENG | 6608 k | 90 % | 92 % | 102 574 |
| de | DEU | 2765 k | 91 % | 93 % | 16 141 |
| nl | NLD | 2114 k | 68 % | 66 % | 3521 |
| da | DAN | 289 k | 63 % | 64 % | 711 |
| is | ISL | 56 k | 54 % | 79 % | 118 |

Table 8: **Wikipedia statistics.** 'Manual edits' include the proportion of edits (of content pages) performed by registered non-bot users or anonymous editors (out of the total number of content page edits performed by anyone, including bots). The number of monthly editors is the mean number of registered non-bot users who edited at least one content page, per month. English, German, Dutch (NLD), Danish (DAN) and Icelandic (ISL) are included for comparison. The wikis with a pin symbol 📍 contain (some) articles tagged by dialect; see footnote 25. *The *nds* and *nds-nl* wikis are primarily concerned with varieties of Low Saxon spoken in, respectively, Germany and the Netherlands. The former also contains articles written in varieties associated with the ISO 639-3 codes WEP and FRS, and the latter with ACT, FRS, GOS, DRT (Drents), SDZ (Sallands), STL (Stellingwerfs), TWD and VEL (Veluws).