# OpenReview forum: "A Survey of Corpora for Germanic Low-Resource Languages and Dialects"
_NoDaLiDa/2023/Conference — NoDaLiDa 2023_

### Official Review · Reviewer_45UX · 2023-02-27
**An extensive overview of corpora for more than 20 Germanic low-resource varieties, broken down according to annotation and availability criteria. Impressive work that is extremely useful for anyone working on NLP for Germanic language varieties.**

**Rating:** 9
**Confidence:** 5

**Review:**

This paper provides an extensive overview of available corpora for more than 20 Germanic low-resource varieties. The corpora are classified according to their annotation (morphosyntactic, semantic/parallel) and to other important factors (audio available or not, curated or not). Information about availability, size, detailed variety annotation and transcription styles are given as well. This is very impressive work and will be extremely useful for anyone working on NLP for Germanic language varieties. The authors even propose to share a companion website that could be updated in the future as additional datasets become available.

While the main contribution of the paper corresponds to the five dataset listings, the text is also highly instructive and provides additional information about different spelling and transcription types, potential pitfalls of using uncurated datasets, and activity statistics of Germanic Wikipedias.

Questions and suggestions:
- The delimitation of the field of study feels a bit arbitrary at times: creoles are skipped, but AAVE, which one might argue has developed from a creole, is present. Emigrant varieties (PDC, YID, American Nordic Speech) are present, but varieties spoken outside of Europe as a result of colonization (Afrikaans, German in Namibia, etc.) are not. Sociolects such as Kiezdeutsch are not represented either. A solution could be to focus primarily on varieties currently spoken in Europe, and to move extra-European varieties to a separate table.
- I could imagine that more parallel datasets could be found e.g. on OPUS. If this is out of scope for the present paper, then Table 2 might have to be redefined a bit (e.g. to parallelism among different dialects or something like that).
- §5.2.1: I suppose this refers to unannotated corpora, or corpora that only have fine-grained dialect distinctions. This might be clarified.
- Table 8: Instead of reporting the percentage of manual edits among all edits, wouldn't it be more relevant to give the absolute number of manual edits (per editor)? If a Wikipedia only gets 1 edit per year and it is manual, this will show as 100% but is hardly a sign for an active Wikipedia.

Typos:
- L022: *the* extent
- L1760: wind > enemy (DEU Feind)
- L1768: the words > these words

Additional resources:
- Hovy & Purschke (2018, https://aclanthology.org/D18-1469/ ) present a large dataset of social media posts from Germany, Austria and Switzerland; a considerable amount of the data is in Swiss German and in (Austro-)Bavarian dialects
- Hämäläinen et al. (2020, https://dl.acm.org/doi/10.1145/3423337.3429435 ) work on a Finland-Swedish dataset
- There is an ongoing project on NamDeutsch ( https://www.geisteswissenschaften.fu-berlin.de/en/v/namdeutsch/index.html ) which may or may not fit the bill for this study.
- Lameris & Stymne (2021, https://aclanthology.org/2021.vardial-1.5 ) provide a POS-tagged corpus for Scots.
- The xSID corpus is being extended to GSW in the context of VarDial 2023 ( https://sites.google.com/view/vardial-2023/shared-tasks )
- L690: The delimitation between GSW (Swiss German) and WAE (Walser dialects) is a bit problematic - it is essentially a continuum. Since corpora like ArchiMob, SwissDial or "All Swiss German Dialects Test Set" contain data from Wallis and are just labeled GSW, I would also label the "Walliserdeutsch/RRO" dataset with GSW, and keep the WAE label to Walser varieties spoken in contexts where Italian is the majority language.

**Paper Type:**

Long paper

---

### Official Review · Reviewer_P1ZE · 2023-03-08
**Useful systematic corpora survey**

**Rating:** 7
**Confidence:** 4

**Review:**

The authors have made a survey of NLP corpora for Germanic Low-resource languages (LRLs) and dialects.

First, they state criteria for corpora to be included, like accessibility and quality, and describe how they collected the corpora. Then they group and label the corpora along several axes, e.g., whether the corpora are written or spech-based; what sort of orthography is used; if speech-based, whether they contain phonic or phonemic transcription; whether and how they are morphologicly or syntactically annotated etc.

The work is carried out carefully and the presentation is perspicuous. Getting this on a web-page with links to the corpora will make a useful tool for computational studies of language history and variation, and for developing further tools for LRLs.

The weaker parts of the paper (if this this counts as shortcomings) are that the paper does not show any computational uses of the corpora, and that it does not construct new corpora: it surveys already existing ones.


**Paper Type:**

Long paper

---

### Official Review · Reviewer_hsdc · 2023-03-09
**clear survey**

**Rating:** 8
**Confidence:** 4

**Review:**

The scope of this paper is to provide a survey of the available datasets for low-resource German languages and dialects. In general, the paper is well written and helpful and the paper achieves this goal well, reporting corpora with 4 subclasses of annotation: 1) morphosyntactic, 2) semantic, 3) dialect-classification, and 4) uncurated. The authors also include a few recommendations in Section 6.1, although it would strengthen the paper if the authors could expand these.



**Paper Type:**

Long paper

---

### Decision · Program_Chairs · 2023-03-17

Accept